# VQR: Automated Software Vulnerability Repair Through Vulnerability Queries

## Abstract

Recently, automated vulnerability repair (AVR) approaches have been widely adopted to combat the increasing number of software security issues. In particular, transformer-based models achieve competitive results. While existing models are learned to generate vulnerability repairs, existing AVR models lack a mechanism to provide their models with the precise location of vulnerable code (i.e., models may generate repairs for the non-vulnerable areas). To address this problem, we base our framework on the vision transformer(VIT)-based approaches for object detection that learn to locate bounding boxes via the cross-matching between object queries and image patches. We cross-match vulnerability queries and their corresponding vulnerable code areas through the cross-attention mechanism to generate more accurate repairs. To strengthen our cross-matching, we propose to learn a novel vulnerability query mask that greatly focuses on vulnerable code areas and integrate it into the cross-attention. Moreover, we also incorporate the vulnerability query mask into the self-attention to learn embeddings that emphasize the vulnerable areas of a program. Through an extensive evaluation using the real-world 5,417 vulnerabilities, our approach outperforms all of the baseline methods by 2.68%-32.33%. The training code and pre-trained models are available at `https://github.com/AVR-VQR/VQR`.

## 1 Introduction

Software vulnerabilities are security flaws, glitches, or weaknesses found in software code that could lead to a severe system crash or be leveraged as a threat source by attackers (CSRC, 2020). According to National Vulnerability Database (NVD), the number of vulnerabilities discovered yearly has increased from 6,447 in 2016 to 20,156 in 2021 and 18,017 vulnerabilities have been found in 2022. This trend indicates more vulnerabilities are being discovered and released every year, meaning that there will be more workloads for security analysts to track down and patch those vulnerabilities. In particular, it may take 58 days on average to fix a vulnerability based on vulnerability statistics reported in 2022 (Edgescan, 2022). Recently, Deep Learning (DL)-based approaches have been proposed to automate the vulnerability repair process by learning the representation of vulnerable programs and generating repair patches accordingly, which may potentially accelerate manual security analysis processes. Specifically, the transformer architecture has been widely adopted to generate accurate vulnerability patches that repair the vulnerable code automatically (Chen et al., 2022; Chi et al., 2022; Berabi et al., 2021; Fu et al., 2022). The attention-based transformer is shown to be more effective than RNNs because its self-attention mechanism learns global dependencies when scanning through each word embedding rather than processing input sequentially.

For the software vulnerability repair (SVR) problem, awareness and attention to the vulnerable code areas including vulnerable statements are crucially important. This further helps to guide an SVR model to emphasize and focus more on the vulnerable statements for producing better repairs. However, it is challenging because the vulnerable areas locate spatially in a source code. Toward this challenge, we observe that object detection in computer vision intuitively shares a similar concept to vulnerability repair because both approaches need to localize specific items in the input. Particularly, by linking the vulnerable code areas in a source code to the objects in an image, we hope to borrow the principles from the VIT-based objection detection approaches (Carion et al., 2020; Zhu et al., 2020; Wang et al., 2021a) to propose a novel solution for the SVR problem.

Figure 1: Intuitively, not all code tokens in a program need to be repaired and the repair can be in multiple areas. Similarly, not all pixels in an image has objects and the objects can appear in multiple locations in an image. Thus, in object detection, object queries are used in VIT-based approaches (Carion et al., 2020; Zhu et al., 2020; Wang et al., 2021a) to predict bounding boxes and locate objects. With a similar principle of object detection, we leverage vulnerability queries to attend more to the vulnerable code tokens in the vulnerable code areas and generate repairs for them.

Specifically, our approach is inspired by the VIT-based approaches for object detection (Carion et al., 2020; Zhu et al., 2020; Wang et al., 2021a) where we connect detecting spatial objects in an image for predicting bounding boxes to localizing vulnerable code tokens in a source code for generating the repair tokens. Our model consists of a vulnerability repair encoder to produce code token embeddings for code tokens and a vulnerability repair decoder to generate repair tokens. Similar to the object queries in the VIT-based approaches for object detection aiming to attend to objects in an image for predicting the corresponding bounding boxes, we devise vulnerability queries (VQ) aiming to attend to the vulnerable areas in a source code for predicting repair tokens. Additionally, the cross-attention mechanism employed in the vulnerability repair decoder assists the VQs in cross-matching and paying more attention to the vulnerable code areas.

Furthermore, for real-world software vulnerabilities, not all code tokens in a source code are considered vulnerable, meaning that only some of the code tokens are likely to be more vulnerable than the others (Nguyen et al., 2021; Fu & Tantithamthavorn, 2022). To strengthen the attention of the VQs to the vulnerable code areas or code tokens, we train an additional model to learn a vulnerability mask. Specifically, given a source code, the vulnerability mask has significantly higher vulnerability scores for the vulnerable code tokens. We then apply the vulnerability mask to both the vulnerability repair encoder/decoder to enrich our approach, named *Vulnerability Query based Software Vulnerability Repair* (VQR). Finally, Figure 1 presents a conceptual overview of our VQR.

In summary, our contributions are **(i)** a novel vulnerability repair framework based on object detection that uses vulnerability queries to generate repair patches; **(ii)** a novel vulnerability query mask that facilitates the repair model to locate vulnerable code tokens more accurately during vulnerability query; **(iii)** a comprehensive evaluation of our proposed approach against other automated vulnerability repair approaches using a benchmark dataset including real-world vulnerabilities.

## 2   RELATED WORK

**Automated Vulnerability Repair (AVR)** is a task that uses machine learning models to generate repair patches for vulnerable programs. RNN-based models such as SequenceR (Chen et al., 2019) have been proposed to encode the vulnerable programs and decode corresponding repairs sequentially. SequenceR used Bi-LSTMs as encoders with unidirectional LSTMs to generate repairs. Recently, attention-based Transformer models have been leveraged in the AVR domain, which was shown to be more accurate than RNNs. For instance, VRepair (Chen et al., 2022) relied on an encoder-decoder Transformer with transfer learning using the bug-fix data to boost the performance of the vulnerability repair on C/C++ programs. SeqTrans (Chi et al., 2022) constructed code sequences by considering data flow dependencies of programs and leveraged an identical architecture as VRepair. On the other hand, Berabi et al. (2021) proposed to use a T5 model pre-trained on natural language corpus (i.e., T5-large (Raffel et al., 2020)) to fix JavaScript programs and Fu et al. (2022) utilized a T5 model pre-trained on source code (i.e., CodeT5 (Wang et al., 2021b)) to repair C/C++ programs. Additionally, Mashhadi & Hemmati (2021) applied the CodeBERT (Feng et al., 2020) model to repair Java bugs. Those large pre-trained language models have demonstrated

strong improvement over RNNs and non-pretrained transformers because the pre-training steps help the models gain better initial weights for the vulnerability repair downstream task than training from scratch. DLFix (Li et al., 2020) and CURE (Jiang et al., 2021) were proposed to generate vulnerability repairs that satisfy test cases. Thus, complete repaired functions are required to train and evaluate models and the problem statement is different from ours described in Section 3.1. Different from the sequence-based methods mentioned above, Dinella et al. (2020) proposed to learn the graph transformation based on the Abstract Syntax Tree (AST) of source code, which used GNNs to represent the program and LSTMs to generate repairs for JavaScript programs.

| Vulnerable Function — CWE-787 (Out-of-bounds Write) | | | Repaired Function | | |
|---|---|---|---|---|---|
| 41 | 41 | GPMF_ERR IsValidSize(GPMF_stream *ms, uint32_t size) | 41 | 41 | GPMF_ERR IsValidSize(GPMF_stream *ms, uint32_t size) |
| 42 | 42 | { | 42 | 42 | { |
| 43 | 43 | if (ms) | 43 | 43 | if (ms) |
| 44 | 44 | { | 44 | 44 | { |
| - | 45 | int32_t nestsize = (int32_t)ms->nest_size[ms->nest_level]; | + | 45 | uint32_t nestsize = (uint32_t)ms->nest_size[ms->nest_level]; |
| 46 | 46 | if (nestsize == 0 && ms->nest_level == 0) | 46 | 46 | if (nestsize == 0 && ms->nest_level == 0) |
| 47 | 47 | nestsize = ms->buffer_size_longs; | 47 | 47 | nestsize = ms->buffer_size_longs; |
| 50 | 50 | } | 50 | 50 | } |
| 51 | 51 | return GPMF_ERROR_BAD_STRUCTURE; | 51 | 51 | return GPMF_ERROR_BAD_STRUCTURE; |
| 52 | 52 | } | 52 | 52 | } |
| Subword-tokens of the vulnerable function $x_i = [t_1, \ldots, t_n]$ | | | Subword-tokens of the vulnerability repair $y_i = [r_1, \ldots, r_k]$ | | |
| ['GP', 'MF', '_', 'ERR', 'IsValid', 'Size', '(', 'GP', 'MF', '_', 'stream', '*', 'ms', ',', 'uint', '32', '_', 't', 'size', ')', '{', 'if', '(', 'ms', ')', '{', 'int', '32', '_', 't', 'nest', 'size', '=', '(', 'int', '32', '_', 't', ')', 'ms', '->', 'nest', '_', 'size', '[', 'ms', '->', 'nest', '_', 'level', ']', ';', 'if', '(', 'nest', 'size', '==', '0', '&&', 'ms', '->', 'nest', '_', 'level', '==', '0', ')', 'nest', 'size', '=', 'ms', '->', 'buffer', '_', 'size', '_', 'l', 'ongs', ';', 'if', '(', 'size', '+', '2', '<=', 'nest', 'size', ')', 'return', 'GP', 'MF', '_', 'OK', ';', '}', 'return', 'GP', 'MF', '_', 'ERROR', '_', 'BAD', '_', 'STRUCT', 'URE', ';', '}'] | | | ['ms', ')', '{', 'uint', '32', '_', 't', 'nestsize', '=', '(', 'uint', '32', '_', 't', ')', 'ms', '->'] | | |

Figure 2: (CWE-787 Out-of-bounds Write) A real-world example (GoPro, 2019) of vulnerability in a C function is caused by an inappropriate variable type definition, which could lead to serious security breaches or system crashes. The left column presents the vulnerable function where below are sub-word tokens $x_i$ used as input for our repair model. It can be seen that only some of the tokens highlighted in red (i.e., tokens corresponding to Line 45) are vulnerable. The right column presents the corresponding repaired function where below are sub-word tokens $y_i$ as the repair patch output by our repair model.

# 3 OUR PROPOSED APPROACH

## 3.1 PROBLEM STATEMENT

Assuming we have a source code data set consisting of vulnerable source code functions along with corresponding repair patches that repair the vulnerable parts of those functions. We denote the data set as $D = \{(x_1, y_1), ..., (x_N, y_N)\}$, where $x_i$ is a vulnerable function and $y_i$ is its repair patch. Note that each $y_i$ is not a complete function but a patch used to repair the vulnerable part in the corresponding $x_i$ as shown in Figure 2. The mapping between $x_i$ and $y_i$ has been completed by Chen et al. (2022) through parsing the code difference between the vulnerable and the fixed version of the source functions. In this paper, we leverage BPE algorithm (Sennrich et al., 2016) to tokenize $x_i$ and consider $x_i$ as a sequence of code tokens denoted as $x_i = [t_1, t_2, ..., t_n]$ where the code token $t_j, j = 1, ..., n$ could be a clean token or vulnerable token (i.e., the tokens highlighted in red in Figure 2). Similarly, a repair patch $y_i = [r_1, ..., r_k]$ where $y_i$ consists of $k$ number of repair tokens $r_j, j = 1, ..., k$. Each code token $t_j$ and repair token $r_j$ will be embedded into a vector for the model to learn its representation as detailed in Section 3.2. We define this problem as a sequence-to-sequence code generation task with an objective to capture vulnerable code tokens in $x_i$ to generate corresponding repair patch $y_i$.

## 3.2 VULNERABILITY QUERY BASED SOFTWARE VULNERABILITY REPAIR

Our approach is inspired by the VIT-based approaches (Carion et al., 2020; Zhu et al., 2020; Wang et al., 2021a) for object detection where we link detecting spatial objects in an image for predicting

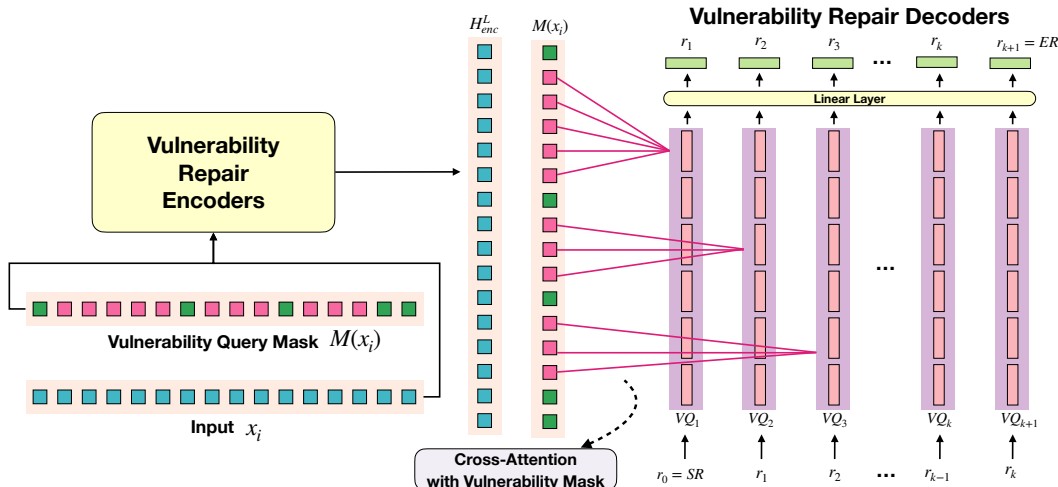

Figure 3: An overview architecture of our VQR approach. Input tokens $x_i = [t_1, ..., t_n]$ and a vulnerability query mask $M(x_i)$ are input to repair encoders that output the embeddings of input tokens $H_{enc}^L$, wherein $M(x_i)$ helps to emphasize the vulnerable embeddings. In repair decoders, each vulnerability query $VQ_i$ is initialized from the previous repair token $r_{i-1}$, which is forwarded through multiple decoder layers followed by a linear layer to generate a repair token $r_i$. In each repair decoder, a cross-attention with $M(x_i)$ to emphasize vulnerable embeddings is leveraged to cross-match $VQ_i$ and $H_{enc}^L$ and generate repairs corresponding to the vulnerable tokens.

bounding boxes to localizing vulnerable code tokens in a source code for generating the repair tokens. Our model consists of an encoder to produce code token embeddings for code tokens and a decoder to generate repair tokens.

Both encoder and decoder are developed based on the transformer architecture (Vaswani et al., 2017). The main component of the encoder is multi-head self-attentions with the aim to learn code token embeddings. Similar to DeTR (Carion et al., 2020), the decoder utilizes both multi-head self-attentions and cross-attentions. The purpose of the cross-attentions is to cross-match vulnerability queries and their corresponding vulnerable code tokens in a vulnerable area. Ideally, when vulnerability queries achieve good matches with their vulnerable code tokens, they possess sufficient information to generate repair tokens.

Additionally, to orient the matching process for attending more to vulnerable code tokens inside a source code, we propose to learn a vulnerability query mask and apply it to both the encoder self-attention and decoder cross-attention mechanism. Particularly, we rely on the information of vulnerable tokens to train an additional model that outputs the possibility of a code token being a vulnerable token. We then base on these vulnerable scores to conduct a vulnerability query mask.

In what follows, we present the technicality of the vulnerability repair encoder, the vulnerability repair decoder, and how to conduct and incorporate vulnerability query masks into our framework.

### 3.2.1 VULNERABILITY REPAIR ENCODER

The purpose of the encoder is to produce code token embeddings for a given source code. Each token is embedded into a vector in $\mathbb{R}^{d=768}$ by an embedding layer and input to the first encoder block. A stack of encoder blocks is leveraged to encode the representation for an embedded sequence through their self-attention layers followed by feed-forward neural networks, and each encoder block can be described as follows:

$$A^t = LN(MultiAttn(H_{enc}^{t-1})) + H_{enc}^{t-1}$$
$$H_{enc}^t = LN(FFN(A^t) + A^t)$$

where the hidden states from the previous encoder block $H_{enc}^{t-1}$ forwards through a multi-head self-attention $MultiAttn$ followed by a 2-layer feed-forward neural network $FFN$, and a layer normal-

ization $LN$. The process will iterate until we obtain the last encoder hidden states $H_{enc}^L$ to represent the vulnerable function. Here we note that $L$ is the number of encoder blocks applied and $H_{enc}^L$ contains the code token embeddings.

### 3.2.2 VULNERABILITY REPAIR DECODER

Input to the vulnerability repair decoder is the vulnerability queries (VQ), each of which aims to match and capture information of vulnerable code tokens in a given source code.

The first VQ embeddings $Q^0 = [q_1^0, ..., q_k^0]$ are conducted and fed through several following decoder blocks. In each block, we apply both multi-head self-attention and cross-attention as follows:

$$\hat{Q}^t = LN(MultiAttn(Q^{t-1})) + Q^{t-1}$$

$$A_{cross}^t = LN(CrossAttn(\hat{Q}^t, H_{enc}^L)) + Q^{t-1}$$

$$Q^t = LN(FFN(A_{cross}^t) + A_{cross}^t)$$

where $H_{enc}^L$ is the encoder output.

It is worth noting that the cross-attention $CrossAttn$ assists us in cross-matching the vulnerability query embeddings $Q^t = [q_0^t, ..., q_k^t]$ and the code token embeddings. If trained appropriately, the vulnerability query embeddings $q_0^t, ..., q_k^t$ attend and emphasize more the vulnerable code token embeddings in the vulnerable function, which finally contain sufficient information to generate the repair tokens.

Eventually, we obtain the output VQ embeddings $Q^U = [q_0^U, ..., q_k^U]$ where $U$ is the number of the decoder blocks applied. On top of these VQ embeddings, we predict the repair tokens $r_1, ..., r_k$. Specifically, we dedicate a linear layer on each VQ embedding $q_0^U, ..., q_k^U$ and aim to predict $r_1, ..., r_k$ and $r_{k+1} = ER$ (i.e., the end repair token) by maximizing the likelihood with respect to a mini-batch of $x_i$:

$$p(y_i \mid x_i) = p(r_1, ..., r_k \mid t_1, ..., t_n) = \prod_{j=0}^{k} p(r_{j+1} \mid q_j^U) \tag{1}$$

where $x_i = [t_1, ..., t_n]$ is the source code and $y_i = [r_1, ..., r_k]$ is the corresponding repair patch.

The next arising question is how to initialize the first VQ embeddings $Q^0 = [q_0^0, ..., q_k^0]$. Different from VIT-based object detection approaches (Carion et al., 2020; Zhu et al., 2020; Wang et al., 2021a), we do not initialize the first VQ embeddings $Q^0 = [q_0^0, ..., q_k^0]$ randomly. Indeed, we initialize $Q^0 = [q_0^0, ..., q_k^0]$ more informatively by setting $q_0^0 = SR$ (i.e., the specific embedding for the start repairing token), $q_j^0 = r_j, j = 1, ..., k$. By this informative initialization, we cast the vulnerability repair problem to the source code to repair-patch generation task.

The inference process is hence very natural. Given a source code $x_i = [t_1, ..., t_n]$, we pass it through the vulnerability repair encoder to work out the encoder output $H_{enc}^L$. We start with the first VQ embedding $q_0^0 = SR$ and feed to the vulnerability repair decoder to generate the first repair token $r_1$. We then set VQ embedding $q_1 = r_1$ and feed it to the vulnerability repair decoder to generate the second repair token $r_2$. We repeat this process until reaching the $ER$ token.

As mentioned before, the key factor to the success of our approach is how to accurately cross-match between the vulnerability queries and the vulnerable code tokens of a given source code. Currently, we expect that the cross-attention mechanism guided by maximizing the likelihood in Eq. (1) supports us in realizing this. To further strengthen the cross-matching, we learn a vulnerability query mask that highly focuses on the vulnerable code tokens and then apply it to the encoder self-attention and the decoder cross-attention mechanism.

### 3.2.3 LEARNING AND INCORPORATING VULNERABILITY QUERY MASK

In what follows, we present how to learn a vulnerability query mask and then apply it to our model.

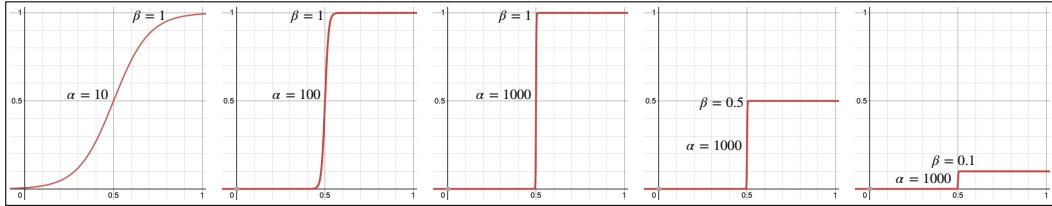

Figure 4: The plots of the vulnerability mask transformation to see how $\alpha, \beta$ control the sharpness. It can be seen that $\alpha$ controls the sharpness (i.e., how fast the curve gets saturated), while $\beta$ controls the gap between vulnerable and non-vulnerable scores.

**Learning vulnerability query mask.** We note that for our dataset $D = \big\{(x_1, y_1), ..., (x_N, y_N)\big\}$, each vulnerable function $x_i = [t_1, ..., t_n]$ is a sequence of code token in which we know exactly the vulnerable scope or information if a code token $t_j$ belongs to a vulnerable statement. In other words, we also possess the token-level vulnerable label $v_i = [u_1, ..., u_n]$ wherein $u_j = 1$ means that the code token $t_j$ belongs to a vulnerable statement and otherwise. For example, in the source code presented in Figure 2, the code tokens highlighted in red are the vulnerable code tokens labelled 1.

We now take advantage of this crucial information to learn vulnerability query masks. Basically, we train an additional model to predict the vulnerability query masks. Specifically, we leverage a pre-trained CodeBERT (Feng et al., 2020) model in learning the vulnerability query masks. Each $t_i$ in $x_i$ is embedded into a vector in $\mathbb{R}^{d=768}$ and forwarded through 12 layers of the BERT architecture. We then use a global max pooling layer and a sigmoid activation to obtain the probability mask $m(x_i)$ and minimize the following cross-entropy loss with respect to a mini-batch of $x_i$

$$H(x_i, v_i) = -\sum_{j=1}^{n} \big[u_j \log m_j(x_i) + (1 - u_j) \log (1 - m_j(x_i))\big] \tag{2}$$

Finally, to sharpen the vulnerability query mask, we apply the following transformation

$$M(x_i) = \frac{\beta}{1 + \exp\{-\alpha(m(x_i) - 0.5)\}}$$

where $\alpha > 0$ and $\beta > 0$ are two parameters to control the sharpness of the vulnerability query mask.

In Figure 4, we visualize how $\alpha$ and $\beta$ affect the vulnerability query masks. It can be seen that $\alpha$ controls the sharpness (i.e., how fast the curve gets saturated), while $\beta$ controls the gap between vulnerable and non-vulnerable scores.

**Applying vulnerability query mask to our model.** We incorporate our vulnerability query mask (VQM) into both encoder output and the cross-attention. For the encoder, we apply as follows:

$$A^t = LN\big(MultiAttn(H_{enc}^{t-1}) + M(x_i) \otimes MultiAttn(H_{enc}^{t-1})\big) + H_{enc}^{t-1}$$

$$H_{enc}^t = LN(FFN(A^t) + A^t)$$

where $\otimes$ is the element-wise product which returns $[M_j(x_i) B_j^t]_{j=1}^n$ with $B^t = MultiAttn(H_{enc}^{t-1})$.

For the cross-attention in the decoder, we apply as follows:

$$\hat{Q}^t = LN(MultiAttn(Q^{t-1})) + Q^{t-1}$$

$$A_{cross}^t = LN\big(CrossAttn(\hat{Q}^t, H_{enc}^L + M(x_i) \otimes H_{enc}^L)\big) + Q^{t-1}$$

$$Q^t = LN(FFN(A_{cross}^t) + A_{cross}^t)$$

Finally, the entire framework of our approach encapsulated the vulnerability repair encoder, vulnerability repair decoder, and how to incorporate vulnerability masks are summarized in Figure 3.

## 4 EXPERIMENTS

We compare our proposed method VQR with existing baseline approaches introduced in Appendix A.2.

### 4.1 EXPERIMENTAL DATASET

We use the same experimental dataset provided by Chen et al. (2022) to evaluate our approach. The dataset consists of Big-Vul (Fan et al., 2020) and CVEfixes (Bhandari et al., 2021) vulnerability fix corpus written in C/C++. The Big-Vul dataset was collected from 348 open-source GitHub projects by crawling the Common Vulnerabilities and Exposures (CVE) database. In total, Big-Vul contains 3,754 code vulnerabilities from 2002 to 2019. On the other hand, the CVEfixes dataset was constructed similarly to the Big-Vul, which consists of 5,365 vulnerabilities collected from 1,754 projects from 1999 to 2021. Specifically, we leverage both datasets pre-processed by Chen et al. (2022) and obtain 5,417 samples spanning 2,095 different vulnerabilities (i.e., CVE-ID) after dropping null and duplicate samples.

### 4.2 PARAMETER SETTING

We split the data into 70% for training, 10% for validation, and 20% for testing. We use a pre-trained T5 model provided by Wang et al. (2021b), which was pre-trained using multiple denoising objectives related to programming languages. Details of the hyperparameter settings for our method during both pre-training and fine-tuning are in Appendix A.1.

### 4.3 MODEL TRAINING

Given that the existing vulnerability repair dataset only contains limited samples, pre-training on a larger bug fix dataset can further enhance the performance of a vulnerability repair model as demonstrated by Chen et al. (2022). The intuition is that the software vulnerability is a sub-domain of the software defect (i.e., bugs) domain which increases the transferability between the two tasks. Thus, for each model including ours, we first pre-train on the bug fix dataset provided by Chen et al. (2022), which consists of 23,607 samples to obtain more meaningful pre-trained weights for the vulnerability repair downstream task. Note that the bug fix dataset is not overlapping with our experimental dataset introduced in Section 4.1. We train our model through specific epochs as reported in Appendix A.1 and select the best model based on the lowest CE loss on the validation set. We run our experiments on a Linux machine with an AMD Ryzen 9 5950X, 64 GB of RAM, and an NVIDIA RTX3090 GPU.

### 4.4 EXPERIMENTAL RESULTS

#### 4.4.1 MAIN RESULTS

We conduct our experiment five times by setting different random seeds and using the dataset described in Section 4.1 and compare our proposed method with the baselines introduced in Appendix A.2. We leverage the percentage of perfect prediction (%PP) evaluation measure as used by Chen et al. (2022) and Fu et al. (2022). %PP is computed as the total correct predictions divided by the total testing samples, specifically, a prediction is considered correct if any of the beam search outputs is exactly the same as the ground-truth repair. During beam search, we use $beam \in [1, 3, 5]$ to evaluate all of the methods. Such beam settings lead to fewer repair candidates generated by the models, which would be more practical in real-world scenarios so developers will not need to inspect many repair candidates. Additionally, we compute BLEU (Papineni et al., 2002) and Meteor (Banerjee & Lavie, 2005) score to measure the similarity between predictions and true labels. The experimental results are shown in Table 1. Our method outperforms all baselines regardless of the number of beams. When comparing only the top-1 repair candidates (i.e., $beam = 1$), our VQR is 2.68%-16.92% better than large pre-trained transformer approaches (i.e., VulRepair and TFix) while our method is 23.18%-24.86% better when comparing with BERT-based approaches (i.e., CodeBERT and GraphCodeBERT). These results confirm that our proposed method of cross-matching vulnerability queries with vulnerable code tokens can help the model encode a more mean-

ingful representation for a vulnerable function and decode the corresponding repair more accurately. In addition, we analyze the performance of the approach for each CWE-ID in Appendix A.3

Table 1: (Main results) The comparison between our VQR approach and other baselines. Accuracy is presented in percentage. Beam=k shows the measure of %PP. The BLEU and Meteor are computed based on Beam=5.

| Methods | Beam=1 | Beam=3 | Beam=5 | BLEU | Meteor |
|---|---|---|---|---|---|
| VQR(**Ours**) | **32.33**±**1.12** | **42.72**±**0.86** | **45.14**±**0.86** | **59.11**±**1.28** | **68.45**±**1.2** |
| VulRepair | 29.65±1.27 | 39.85±1.31 | 42.79±1.15 | 58.7±1.26 | 68.1±1.4 |
| TFix | 15.41±1.96 | 26.7±1.69 | 30±1.77 | 50.71±2.07 | 60.04±2.14 |
| GraphCodeBERT | 9.15±0.43 | 16.83±0.85 | 21.38±0.54 | 49.38±0.45 | 64.4±0.36 |
| CodeBERT | 7.47±0.61 | 13.69±0.37 | 16.85±0.17 | 43.56±0.9 | 59.23±1.01 |
| VRepair | 5.36±0.55 | 10.31±0.29 | 13.12±0.53 | 40.06±0.62 | 55.31±0.63 |
| SequenceR | 0.0±0 | 0.44±0.13 | 0.53±0.27 | 7.39±0.63 | 27.74±1.6 |

### 4.4.2 ABLATION STUDY

**(1) Study the effectiveness of our proposed vulnerability query and mask.** To this end, we compare our proposed method with other variants as follows:

- **Perfect Vulnerability Masking in Encoders and Decoders**: This method uses an identical architecture as our VQR, however, the perfect vulnerability masks (i.e., the exact location of each vulnerable token) are provided instead of predicted by a localization model.

- **Vulnerability Query Masking in Encoders**: This method only applies vulnerability query masks on the self-attention output of each encoder to help the model focus more on vulnerable tokens when encoding the representations for a vulnerable function.

- **Vulnerability Query Masking in Decoders**: This method only applies vulnerability query masks on the decoder cross-attention when cross-matching vulnerability queries and vulnerable code tokens to support the model to focus more on vulnerable tokens when generating repair tokens.

- **Without Vulnerability Query Masking**: This method is a plain transformer encoder-decoder architecture that applies no vulnerability query mask.

- **With Vulnerability Query Randomly Initialized**: This method applies vulnerability query masks in both encoders and decoders while vulnerability queries are randomly initialized at the start of training.

The experimental results are shown in Table 2. It can be seen that our approach to initialize the vulnerability query (VQ) based on repair tokens during training consistently outperforms the randomly initialized VQ. However, the random VQ method still outperforms baselines such as CodeBERT and GraphCodeBERT, highlighting the effectiveness of using vulnerability queries with cross-attention (as proposed in Section 3.2.2) for our vulnerability repair task. Moreover, applying the vulnerability query mask is beneficial for both encoder self-attention and decoder cross-attention. It enhances the %PP by 2.93% when applied to encoders while gaining a %PP of 3.02% when applied to decoders. While the vulnerability query mask benefits both encoders and decoders, our proposed method to leverage the mask on both sides achieves better results for $beam \in [1, 3]$. These results confirm that using our mask on both encoders and decoders is more beneficial than using it on either side. Last but not least, the perfect vulnerability query mask achieves the highest %PP, highlighting the effectiveness of our vulnerability query mask.

To further demonstrate that similar to the capability of object queries in capturing information of bounding objects in images, vulnerability queries (VQ) are capable of attending to and capturing information of vulnerable scopes in source codes. We visualize the cross-attention map between VQs (x-axis) and all token representations of source code (y-axis) in Figure 5. As we expect, the VQs attend to vulnerable scopes, hence capturing sufficient information for predicting the repair patches. More visualizations can be found in Appendix A.4

Table 2: (Ablation results) The comparison between our proposed method and four other variants. Accuracy is presented in percentage.

| Methods | Beam=1 | Beam=3 | Beam=5 |
|---|---|---|---|
| Perfect Mask Encoder + Perfect Mask Decoder | *33.76* | *44.31* | *46.88* |
| Vul Mask Encoder + Vul Mask Decoder (ours) | **33.21** | **44.04** | 46.06 |
| Vul Mask Encoder | 32.75 | 43.49 | **46.15** |
| Vul Mask Decoder | 32.84 | 43.85 | 45.69 |
| w/o Vul Mask | 29.82 | 39.72 | 43.67 |
| with Vul Query randomly initialized | 12.57 | 24.95 | 28.81 |

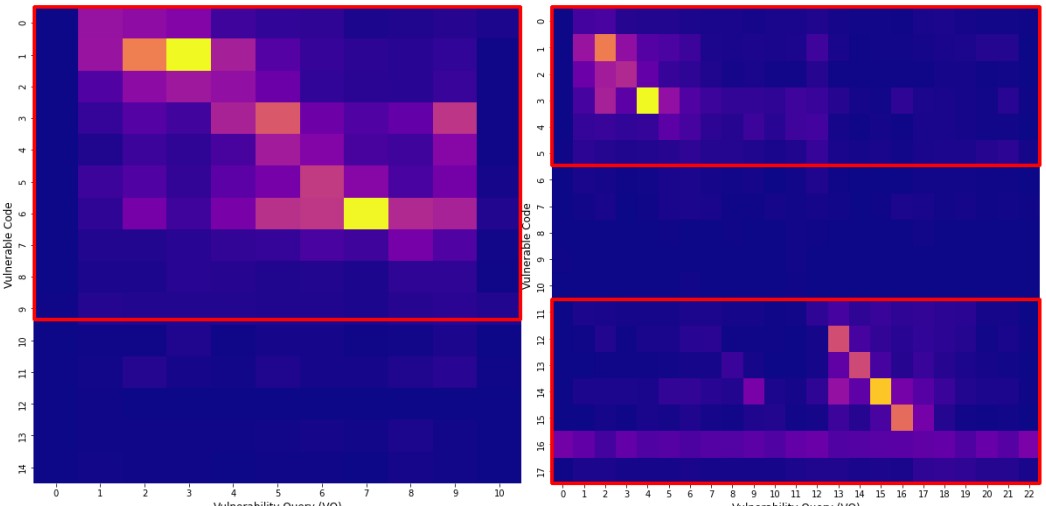

Figure 5: The visualization of the cross-attention scores between vulnerability queries (axis X) and vulnerable code representation (axis Y) in two vulnerable functions. The vulnerable scopes are highlighted in red boxes. It can be seen that the attention scores are highly activated between the interaction of vulnerable tokens representation and vulnerability queries.

**(2) Study the effectiveness of pre-training on bug fix corpus.** We aim to study whether pre-training on a larger bug fix corpus support AVR models to perform better vulnerability repairs. To this end, we directly train each baseline approach and our VQR on the vulnerability repair data without pre-training on the bug fix data. The results shown in Table 3 correspond to the finding by Chen et al. (2022) that the knowledge from the general bug fix corpus can be transferred to benefit the performance of AVR models. Our method still outperforms all baseline approaches.

Table 3: Compare our method with baselines, where all the methods are not pre-trained on the bug-fix data Chen et al. (2022). Accuracy is presented in percentage.

| Methods | Beam=1 | Beam=3 | Beam=5 |
|---|---|---|---|
| VQR | **5.32** | **8.81** | **9.72** |
| VulRepair | 4.13 | 6.06 | 7.43 |
| TFix | 2.75 | 4.4 | 4.68 |
| GraphCodeBERT | 2.57 | 4.13 | 5.23 |
| CodeBERT | 1.56 | 2.29 | 2.75 |
| VRepair | 0.09 | 0.55 | 0.92 |
| SequenceR | 0 | 0 | 0 |

## 5 CONCLUSION

In this paper, we have introduced a new AVR method inspired by VIT-based approaches for object detection to enhance awareness and attention to vulnerable code areas in a vulnerable function for producing better repairs. In our repair model, we cross-match vulnerability queries and their corresponding vulnerable code areas and their corresponding repairs via the cross-attention mechanism. To strengthen such cross-matchings, we propose to learn a vulnerability query mask that highly focuses on vulnerable code areas and incorporate it into the cross-attention. Additionally, we also apply the vulnerability query mask in the self-attention of encoders to help our model focus more on vulnerable code tokens when learning the embeddings of each token. Through an extensive evaluation of 5,417 real-world vulnerabilities, our approach outperforms all of the baseline approaches.

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

# A    APPENDIX

## A.1    TRAINING SCHEME OF OUR VQR APPROACH

Table 4: Training scheme of our VQR approach. Note. #: Scheme for training the mask prediction model; *: Scheme for training the repair model.

| Training | Data | $Seq_{enc}$ | $Seq_{dec}$ | Optim | Sch. | LR | Grad Clip | Bhz | Epo |
|---|---|---|---|---|---|---|---|---|---|
| #Pre-train | Bug Fix | 512 | N/A | AdamW | Linear | 1e-4 | 1.0 | 16 | 75 |
| #Fine-tune | Vul Fix | 512 | N/A | AdamW | Linear | 1e-4 | 1.0 | 16 | 75 |
| *Pre-train | Bug Fix | 512 | 256 | AdamW | Linear | 1e-4 | 1.0 | 8 | 75 |
| *Fine-tune | Vul Fix | 512 | 256 | AdamW | Linear | 1e-4 | 1.0 | 8 | 75 |

## A.2    BASELINE APPROACHES

We compare our proposed method VQR with the following baselines:

- **VRepair**: The vanilla transformer architecture is first pre-trained on a bug-fixing corpus in a supervised manner, VRepair (Chen et al., 2022) is then fine-tuned to fulfil the vulnerability repair task which shares a similar nature to the bug-fixing task. We reproduce VRepair by following the instruction provided by Chen *et al.* to build and train the model.

- **VulRepair**: The transformer encoder-decoder approach learns the representation of vulnerable functions and generates the corresponding repair patches. VulRepair (Fu et al., 2022) relies on the CodeT5 (Wang et al., 2021b) model pre-trained through multi-task learning on the Codesearchnet (Husain et al., 2019) dataset to learn the characteristics of programming languages. We reproduce VulRepair using the repository provided by Fu *et al.*.

- **TFix**: Similar to VulRepair (Fu et al., 2022), TFix (Berabi et al., 2021) relies on a transformer encoder-decoder model, however, pre-trained on natural language using denoising objectives (Raffel et al., 2020) (i.e., T5-large). We reproduce TFix using the repository provided by Berabi *et al.*.

- **SequenceR**: An RNN-based approach that learns the representation of vulnerable functions token by token, SequenceR (Chen et al., 2019) leverages bi-directional LSTM encoders with unidirectional LSTM decoders. We reproduce SequenceR by following the instruction provided by Chen *et al.* to build and train the model.

- **CodeBERT**: A bi-directional self-attention pre-trained on the Codesearchnet (Husain et al., 2019) dataset using masked language modelling and replaced token detection objectives (Feng et al., 2020). Mashhadi & Hemmati (2021) leveraged CodeBERT (Feng et al., 2020) for automated program repair of Java bugs and presented substantial improvement over RNN-based models. We reproduce CodeBERT using the repository provided by Feng *et al.*.

- **GraphCodeBERT**: An extensive version of CodeBERT (Feng et al., 2020), GraphCodeBERT (Guo et al., 2021) considers the Data Flow Graph (DFG) of code using the graph-guided masked attention during both pre-training and fine-tuning stages. We reproduce GraphCodeBERT using the repository provided by Guo *et al.*.

Note we follow the best hyperparameter setting as reported by the original authors to obtain the best results of each baseline method.

## A.3    ANALYSIS OF OUR VQR'S PERFORMANCE

We visualize the %PP across all CWE-IDs in our testing data as a bar graph to explore our VQR's performance for different CWE-IDs. In addition, we show the frequency of each CWE-ID for both training and testing data as two line graphs to explore the relationship between the frequency of samples and the performance of our method. Note that the ticks of the Y axis on the left are for the %PP metric while those on the right are for the data frequency of each CWE-ID.

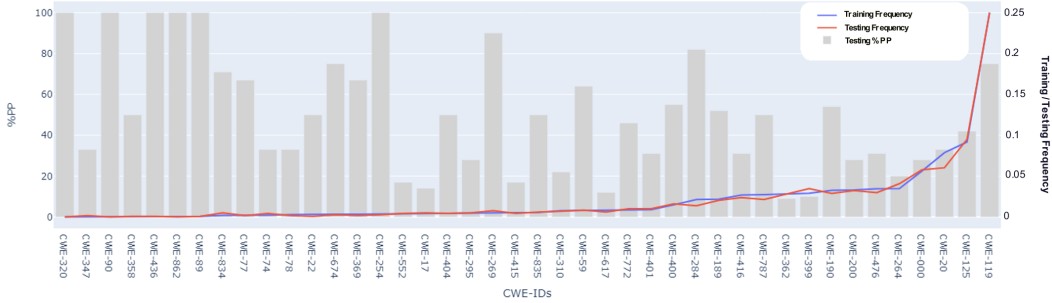

Figure 6: The performance analysis of our VQR based on different vulnerability types (i.e., CWE-IDs). The bar chart represents the %PP while the blue line is the training frequency and the red line is the testing frequency across all vulnerability types. Note that the ticks of the Y axis on the left are for the %PP metric while those on the right are for the data frequency of each CWE-ID.

As shown in Figure 6, the performance of our approach varies for each CWE-ID. Our approach performs well on some of the CWE-IDs that all testing samples can be correctly repaired. Furthermore, we find that the frequency of training and testing samples are not highly correlated with the performance of our method. This indicates that automated vulnerability repair (AVR) is a challenging problem in that high-frequency samples may not guarantee the repair model's performance.

Table 5: The %PP of our VQR approach across the top 25 most dangerous CWE-IDs in 2022. The %PP is shown based on the beam search results where Beam=5.

| Rank | ID | %PP |
|------|---------|-----------------|
| 1 | CWE-787 | 50% (13/26) |
| 3 | CWE-89 | 100% (2/2) |
| 4 | CWE-20 | 33% (24/72) |
| 5 | CWE-125 | 42% (48/113) |
| 6 | CWE-78 | 33% (1/3) |
| 7 | CWE-416 | 31% (9/29) |
| 8 | CWE-22 | 50% (1/2) |
| 11 | CWE-476 | 31% (11/36) |
| 13 | CWE-190 | 54% (19/35) |
| 16 | CWE-862 | 100% (1/1) |
| 17 | CWE-77 | 67% (2/3) |
| 19 | CWE-119 | 75% (223/296) |
| 22 | CWE-362 | 9% (3/34) |
| 23 | CWE-400 | 55% (11/20) |
| | Average | 55% (368/672) |

To investigate whether our VQR approach can repair dangerous real-world vulnerabilities, we evaluate our approach based on the 2022 CWE Top-25 Most Dangerous Software Weaknesses released by the CWE community at `https://cwe.mitre.org/top25/archive/2022/2022_cwe_top25.html`. The results are presented in the above table. We find that our approach can correctly repair 55% of the vulnerable functions affected by the Top-25 most dangerous CWE-IDs, which is better than the average performance of our approach (i.e., 45.14%).

## A.4 ANALYSIS OF OUR VQR'S VULNERABILITY QUERY AND MASK

To demonstrate that our vulnerability query (VQ) and vulnerability mask can learn better cross-attention. We visualize the cross-attention scores between vulnerability queries and vulnerable source code representation. We compare the visualization for three approaches as follows:

- **Vul Query + Vul Mask (Our VQR method)**: Our proposed method that utilizes both vulnerability query (initialized based on repair tokens) and vulnerability mask.
- **Random Vul Query**: The method that utilizes randomly initialized vulnerability query and vulnerability mask.
- **Without Vul Mask**: The method that utilizes vulnerability query (initialized based on repair tokens) without using vulnerability mask.

In the example with one vulnerable code area shown in Figure 7, the cross-attention score of our method has warmer colour in the vulnerable area highlighted by the red box. Thus, our method activates the cross-attention score between the vulnerable code representation and their corresponding vulnerability queries more than the other two variant approaches in the vulnerable area.

In the example with seven vulnerable code areas shown in Figure 8, the cross-attention score of our method has warmer colour in the vulnerable areas highlighted by the red box. Furthermore, our method has better contrast than the other two methods, in that the cross-attention scores between vulnerable code representation and their corresponding VQs are highly activated while the cross-attention scores between vulnerable code representation and non-corresponding VQs are slightly activated.

These results indicate that our proposed VQ with the vulnerability mask method can guide vulnerability queries to attend to their corresponding vulnerable code areas.

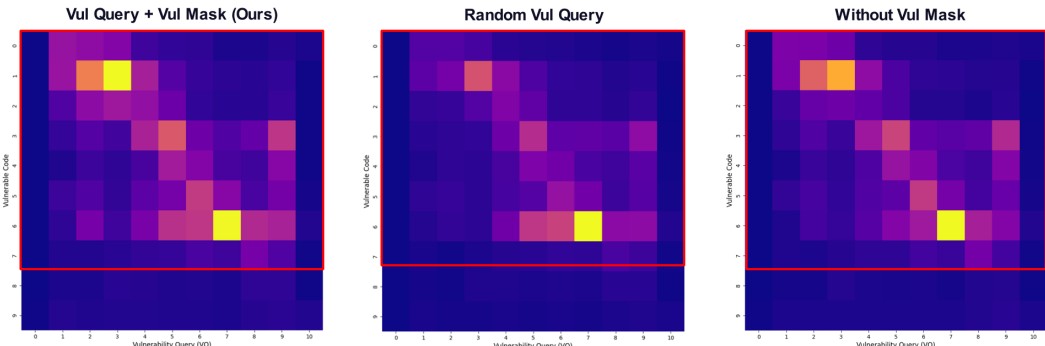

Figure 7: The visualization of the cross-attention scores between vulnerability queries (axis X) and vulnerable source code representation (axis Y). The vulnerable scopes are highlighted in red boxes. Our method on the left is warmer than the other two, which activates the most cross-attention scores between vulnerable code representation and vulnerability query in the vulnerable scopes.

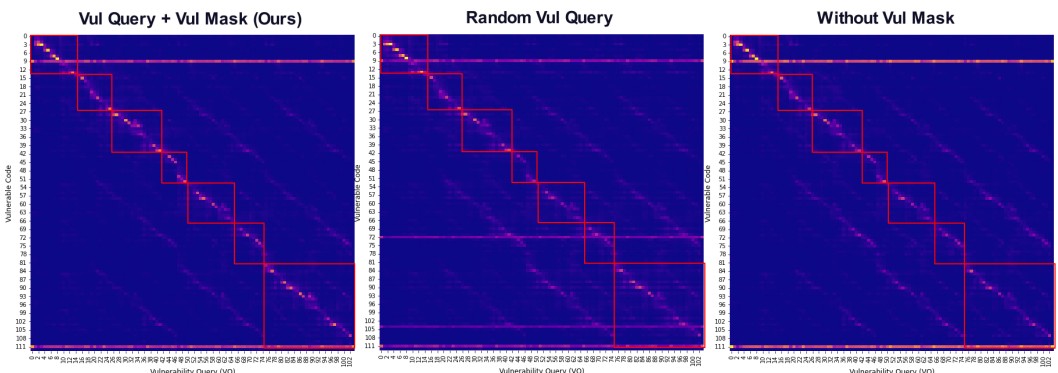

Figure 8: The visualization of the cross-attention scores between vulnerability queries (axis X) and vulnerable source code representation (axis Y). The vulnerable scopes are highlighted in red boxes. Our method on the left has the best contrast between cross-attention scores in vulnerable code areas and non-vulnerable code areas. In other words, the cross-attention activated by our method in the vulnerable scopes is higher than the other two methods while the cross-attention scores are lower than the other two methods when outside vulnerable scopes.

