# OpenReview forum: "VQR: Automated Software Vulnerability Repair Through Vulnerability Queries"
_ICLR.cc/2023/Conference — Submitted to ICLR 2023_

### Official Review · Reviewer_oYMK · 2022-10-23

**Confidence:** 4
**Correctness:** 2
**Technical Novelty And Significance:** 3
**Empirical Novelty And Significance:** 3
**Recommendation:** 5

**Clarity, Quality, Novelty And Reproducibility:**

### Clarity

The paper is very well written.

### Quality

The paper is of very high quality.

### Novelty

The main idea which is an application of Vision Transformers to code
vulnerability detection is novel.

### Reproducibility

The complete source code is provided.


**Strength And Weaknesses:**

### Strengths

The paper is based on a good idea, is very well written, and presents convincing
results with many comparisons to state-of-the-art and an ablation study.

### Weaknesses and Questions

My main concern is that I'm not sure about the analogy between VIT and VQR,
particularly between object queries and vulnerability queries. Object queries
are used by the model to predict objects in a given area of the image, are
learned throughout the training process (as an embedding), and are the same for
each image (apart from being updated by learning). In contrast, it seems to me
based on 3.2.2 that vulnerability queries are just inputs to the decoder. This
is reinforced by their initialization: they are initialized by the repair tokens,
which are different for each example (they are the targets).

If I'm not mistaken then some of the parallels between VIT and VQR and some of the
arguments in the paper do not hold, and the state-of-the-art results are
obtained probably because of the architecture from VIT and not the queries,
which makes it necessary to rewrite significant parts of the paper, including
the title.

I believe that a minus sign is missing from the cross-entropy loss (2). Also,
maybe it would be clearer to explicitly state that this is a cross-entropy loss.

$x_i = [t_1, \dots, t_n]$ is sometimes referred to as a sequence of tokens, and
sometimes as a sequence of token embeddings.

It could be helpful to expand the VIT abbreviation in the paper.

There are some small typos:
- in the Abstract, "the real-world 5417 vulnerabilities"
- at the start of 3.1, "Assuming"



**Summary Of The Paper:**

The paper applies a Vision Tranformer (VIT)-based approach to automated
vulnerability repair. The method is called Vulnerability Query based Software
Vulnerability Repair (VQR). The authors introduce vulnerability queries similar
to the object queries of VIT and cross-match them with vulnerable code areas
through a cross-attention mechanism to generate more accurate repairs. They also
learn a vulnerability query mask to focus on vulnerable code areas and include
it into the cross-attention, and also into the self-attention in the encoder to
learn embeddings that emphasize the vulnerable areas of the program. Their
method outperforms existing state-of-the-art methods based on a real-world dataset
of 5417 vulnerabilities.


**Summary Of The Review:**

I really liked the paper, but I have a serious concern about the vulnerability
queries which are at the heart of it. If I'm mistaken regarding this issue it
probably needs to be made clearer in the paper.

---

### Official Review · Reviewer_urCv · 2022-10-24

**Confidence:** 4
**Correctness:** 3
**Technical Novelty And Significance:** 2
**Empirical Novelty And Significance:** 2
**Recommendation:** 6

**Clarity, Quality, Novelty And Reproducibility:**

The paper is written clearly (except for some points above). The authors have given a link their code and models so it should be possible to reproduce the results. The paper makes a simple modification to the transformer architecture and has minimal modeling contribution.

**Strength And Weaknesses:**

Strengths
-----
* The problem of fixing software vulnerabilities is important in practice and paper performs evaluation on two of the most common vulnerability datasets against relevant baselines.
* It achieves a few percentage improvement across different beam sizes.

Weaknesses
----
* The modeling contribution of the paper is minimal. It incorporates a simple masking strategy that labels vulnerable code tokens and uses the masks in the self/cross attention computations.
* The test set is split arbitrarily into 80-20 train+validation and test. The test set is small enough (5.4K samples) that a five fold experiment can be conducted. Can you report results on the entire set of examples similar to Table 2?
* The training setup and accuracy of the vulnerability query mask prediction model are not reported.

Clarifications
--------
* If you are generating token-wise labels as query masks then why and how do you use global pooling (see the sentence before Eq 2)?
* Which layers do you apply the query marks to in the encoder and the decoder?

**Summary Of The Paper:**

This paper proposes a transformer model for generating repair for software vulnerabilities. It trains a separate token-level classifier model to generate masks called vulnerability queries and incorporates these masks in both the encoder and decoder parts. Through experimental evaluation on Big-Vul and CVEFixes, it shows that this helps the model obtain better repairs compared to baseline models.

**Summary Of The Review:**

The paper addresses a specific software engineering problem and makes a simple modification to transformers to encode domain knowledge. On one split of the test data, it gets a few percentage point improvement over baselines.

---

### Official Review · Reviewer_Vf8F · 2022-10-25

**Confidence:** 4
**Correctness:** 3
**Technical Novelty And Significance:** 2
**Empirical Novelty And Significance:** 2
**Recommendation:** 5

**Clarity, Quality, Novelty And Reproducibility:**

Clarity: Overall, the paper is easy to follow.

Novelty: The novelty of this work is incorporating vulnerability query mask that could potentially improve existing transformer based code repair models.

Reproducibility: Model configurations are shared to enable reproducibility of the proposed work.


**Strength And Weaknesses:**

Strength

- Code repair is a challenging task and improving the accuracy of the transformer based models is a reasonable direction.
- Ablation studies are performed to demonstrate the effectiveness of added vulnerability query mask.

Weakness

-  The improvement over the existing transformer based models proposed in this work has limited novelty. Proposed vulnerability is a direct mapping of the vision transformer setup.

- The evaluation results do not discuss types of errors used for training and evaluation. It is also not clear if the base models using text-to-text transformers (e.g., TFix) are fine tuned with the same training dataset. What implementation of CodeBERT is used for the evaluation of vulnerability repair?




**Summary Of The Paper:**

This paper aims at improving the transformer based vulnerable code repair techniques. Inspired by vision transformer based object detection approach, it proposes to incorporate a learnable  ‘vulnerability query mask’ into both encoder output and the cross-attention of existing transformer based models. Experimental results show the proposed approach provides an increased percentage of perfect predictions compared to state-of-the art approaches.

**Summary Of The Review:**

The proposed approach does not demonstrate significant novelty. Also, the effectiveness of the proposed approach should be investigated more elaborately by examining the error types used for training and evaluation.

---

### Official Review · Reviewer_VFto · 2022-10-25

**Confidence:** 4
**Correctness:** 3
**Technical Novelty And Significance:** 3
**Empirical Novelty And Significance:** 3
**Recommendation:** 3

**Clarity, Quality, Novelty And Reproducibility:**

Quality: Weak. The effectiveness of the design is not well demonstrated.
Clarity: Good. The writing of the paper is good and clearly shows the proposed method.
Originality: Good. The work sheds some light on the solution to code repair.


**Strength And Weaknesses:**

Strength:
(1) The research problem is important among code repair community.
(2) The idea of incorporating predicting vulnerable areas into code repair is interesting and the intuition makes sense.


Weakness:
(1) The concept of vulnerability query is somewhat weird as the authors cannot fully elaborate what it exactly is, especially when making analogy to object query. The vulnerability query is designed and utilized in the decoder part, how it functions is also not clearly described.
(2) Some important baselines are missing and more metrics should be reported. There are a handful of works such as [1,2] which also use seq2seq models to predict program repair but these baselines are missing in experiments. Sequence based metrics should also be reported like BLEU, METEOR as they are widely used in previous code repair works.
(3) The ablation study is not sufficient. While the authors highlight the importance of proposed vulnerability query, there is no ablation study to demonstrate the effectiveness of it.

[1] Li, Yi, Shaohua Wang, and Tien N. Nguyen. "Dlfix: Context-based code transformation learning for automated program repair." Proceedings of the ACM/IEEE 42nd International Conference on Software Engineering. 2020.
[2] Jiang, Nan, Thibaud Lutellier, and Lin Tan. "Cure: Code-aware neural machine translation for automatic program repair." 2021 IEEE/ACM 43rd International Conference on Software Engineering (ICSE). IEEE, 2021.


**Summary Of The Paper:**

This paper aims to improve existing automated vulnerability repair problem by considering the prediction of precise parts of vulnerable code. By making an analogy to object detection problem in computer vision, they introduce the concept of vulnerability query counter to object query and adopt cross-attention mechanism to pay attention to vulnerable code areas. Besides, they also propose to learn an extra model to predict vulnerable areas, the prediction is used to construct a vulnerability query mask and further incorporated into both encoder and decoder’s self-attention. Evaluation is conducted on a read-world dataset which achieves outperforming results.

**Summary Of The Review:**

This work presents the key idea of providing code repair model with the precise location of vulnerable code and proposes two techniques: vulnerability query cross-match and vulnerability query mask. But the rationale and effectiveness of vulnerability query is not well explained and proved. And important baselines are also missing during the experimental evaluation. We believe that this paper can be further improved to be solid work.

---

### Decision · Program_Chairs · 2023-01-20

**Decision:**

Reject

**Justification For Why Not Higher Score:**

Remaining questions about robustness of improvements and clarity of contributions.

**Justification For Why Not Lower Score:**

N/A

**Metareview: Summary, Strengths And Weaknesses:**

This paper presents a method for repairing vulnerable source code that is based on an encoder-decoder Transformer architecture, but additional supervision is provided indicating if each input token appears inside a "vulnerable statement" (one that needs repairing). The learned masks are then used to modify the attention operations, with the intuition being that these masks focus attention in a similar way as object queries are used in Vision Transformers. While the authors presented a number of helpful additional experiments in the rebuttal, there remain concerns about the clarity of the contributions. Reviewers are unconvinced that the framing of vulnerability queries is directly analogous to object queries, and the significance of the architectural contribution remains somewhat unclear, as the gains are models and only in one experimental setting, and there is some mixing in several comparisons of contributions from using the pretraining set of Chen et al. with the architectural modifications. To match the narrative being presented, I'd suggest the authors focus on demonstrating the robustness of improvements from just the architectural component, ideally holding fixed the pretraining data and demonstrating improvements across more kinds of repair tasks.